# Co-Occurrence of *Filifactor alocis* with Red Complex Bacteria in Type 2 Diabetes Mellitus Subjects with and without Chronic Periodontitis: A Pilot Study

Hawaabi F. M. Shaikh [1], Pratima U. Oswal [1], Manohar S. Kugaji [2], Sandeep S. Katti [1], Kishore G. Bhat [2] and Vinayak M. Joshi [3,*]

[1] Department of Periodontology, Maratha Mandal's Nathajirao G. Halgekar Institute of Dental Sciences & Research Centre, Belagavi 590010, Karnataka, India

[2] Central Research Laboratory, Maratha Mandal's Nathajirao G. Halgekar Institute of Dental Sciences & Research Centre, Belagavi 590010, Karnataka, India

[3] Department pf Periodontics, School of Dentistry, Louisiana State University Health Science Centre, New Orleans, LA 70119, USA

* Correspondence: vjosh1@lsuhsc.edu

**Abstract:** The periodontal disease etiology has been a demesne of scrupulous research, with a myriad of bacterial phylotypes inhabiting the periodontal pockets. The aim of our study was to assess the frequency of *Filifactor alocis* in type 2 diabetes mellitus (DM) subjects having a healthy periodontium (DH) or chronic periodontitis (DCP) and its correlation with clinical parameters and red complex bacteria. Polymerase chain reaction was carried out for the detection of *F. alocis* and red complex bacteria from subgingival plaque samples. The data were analyzed using Fisher's Exact Test and Pearson's chi-square test. A $p$ value $< 0.05$ was considered statistically significant. *F. alocis* was detected at considerably higher levels in DCP ($p < 0.05$). *F. alocis* presence was also positively correlated with *T. forsythia* detection and the clinical parameters PD and CAL ($p < 0.05$). Subjects with good glycemic control showed a considerably lower detection of *F. alocis* as compared to fair- and poor-glycemic-control subjects. This is the first paper reporting the co-occurrence of *F. alocis* and *T. forsythia* in diabetic subjects with chronic periodontitis. These findings show that *F. alocis* can play an important role in establishing synergistic collaborations with other pathogenic oral microorganisms and speeding up the course of periodontal disease in diabetics.

**Keywords:** diabetes mellitus; chronic periodontitis; subgingival plaque; *Filifactor alocis*; polymerase chain reaction

## 1. Introduction

Periodontitis is a multifactorial infectious disease that leads to the destruction of tooth-supporting structures such as the periodontal ligament, bone and cementum. Its etiology has been a demesne of scrupulous research in the past decades. While periodontal pockets harbor a myriad of bacterial phylotypes, it is extremely toilsome to distinguish between mere commensals and true pathogens. Socransky et al. described different microbial complexes playing a role in the initiation and progression of periodontal disease [1]. Among them, red complex bacteria comprising *Porphyromonas gingivalis*, *Treponema denticola* and *Tannerella forsythia* are known to cause destructive events in periodontitis and contribute to the disease development [2]. Apart from these, the identification of previously uncultivable and fastidious species in periodontitis patients using novel, culture-independent techniques contributed fresh knowledge on bacterial communities in the periodontal pocket [3].

The association between diabetes mellitus and periodontitis has shown to be bidirectional in nature [4]. Diabetes is known to be one of the risk factors for gingivitis and periodontitis which enhances the probability of progressive bone loss and attachment loss

over time. In addition, various studies have shown that the glycemic control of diabetics with periodontitis worsens much more as compared to that of diabetics without periodontitis [5–7]. There is a multifold increase in alveolar bone loss and attachment loss between uncontrolled diabetic and non-diabetic periodontitis patients, implying that changes in the host immunoinflammatory response play a major role in tissue degradation [8]. There is a distinctive biosphere of heterogenous and facultative periodontal microbial species in diabetics, which clusters significantly based on HbA1c levels [9]. *Filifactor alocis*, a fastidious, Gram-positive, asaccharolytic, slow-growing, obligate anaerobic rod, is one of the newly acknowledged quiescent periodontal pathogens having trypsin-like enzymatic activity analogous to that of *P. gingivalis* and *T. denticola* [3,10]. It can also thrive in the periodontal pocket because of its ability to endure oxidative stress [11]. Previous studies have associated *F. alocis* with periodontitis at escalated frequency and numbers [9,10]. It is ascribed as the second most prevailing bacterium in chronic periodontitis (CP) and the third most prevailing in generalized aggressive periodontitis and therefore has been proposed to be an exceptional marker pathogen for periodontal disease [11].

*F. alocis*, due to its slow-growing nature, is difficult to detect by conventional culture-based methodologies. Therefore, in the present study, PCR was used for the detection of *F. alocis* and red complex bacterial species. Multiplex PCR is time-saving as well as reduces the bulk of reagents and templates required. In the present study, *F. alocis* and red complex species could not be analyzed together in multiplex PCR, as they have different annealing temperatures, and their primers are complementary and may bind to each other rather than to the target bacteria. Specific primers for each chosen bacteria will target the designated sequences of the 16S rDNA. The 16S rDNA gene is universally distributed amongst bacteria and has trademarks within its sequence.

Hence, the aim of our present study was to evaluate the prevalence of *F. alocis* and the co-occurrence of *F. alocis* and red complex bacteria (*P. gingivalis*, *T. denticola* and *T. forsythia*) in type 2 diabetes mellitus (DM) subjects having a healthy periodontium and CP.

## 2. Materials and Methods

A total of 98 Type 2 DM subjects, aged 30–70 years, were screened for our study. Subjects reporting to the outpatient department of the Department of Periodontology, Maratha Mandal's Nathajirao G. Halgekar Institute of Dental Sciences and Research Centre, Belagavi, Karnataka, were recruited in the study. A systematic methodology was used to interview all participants. The participants were quizzed on their medical history, medications, smoking habits, periodontitis history and maintenance of oral hygiene. The institutional ethical committee authorized the study procedure, and all subjects gave written informed consent prior to their examination. Moreover, subjects having less than 20 teeth were not included. The inclusion criteria needed that all the subjects recruited for the study were positive for type 2 DM and had minimum 20 teeth. The study subjects were selected based on the history of having been diagnosed with Type 2 DM. Patients with HBA1c $\geq$ 6.0 were selected for the study. The study subjects with Type 2 DM diagnosed with generalized chronic periodontitis were grouped in the diabetic chronic periodontitis group (DCP group), and the subjects with Type 2 DM free from periodontitis were grouped in diabetic healthy (DH group) [12].

The subjects included in the DH group demonstrated no indications of gingival inflammation or bleeding on probing at a probing depth (PD) $\leq$3 mm and showed $\leq$1 mm of clinical attachment loss (CAL). In the DCP group, subjects that demonstrated the presence of gingival inflammation with bleeding on probing at PD $\geq$ 5 mm and CAL $\geq$ 3 mm were included [12].

The criteria for exclusion from the study were: (1) subjects who had received periodontal therapy or antimicrobial therapy in the previous 3 months before sampling, (2) use of oral rinses in the last 3 months, (3) presence of any systemic disease and endocrine disorder other than Type 2 DM, (4) consumption of any form of smoke/smokeless tobacco, (5) pregnant and lactating women, (6) subjects with no HbA1c reports.

### 2.1. Clinical Periodontal Examination

The clinical periodontal examination comprised the assessment of the plaque index (PI), the gingival index (GI) [13], the bleeding index (BI) [14], PD and CAL. The PD and CAL values were recorded at six sites per tooth. The subjects were also assessed for their hemoglobin A1c (HbA1c) values [15]. All throughout the investigation, two examiners performed all measurements with a millimeter-graded probe (UNC 15). An inter-examiner calibration of volunteers was used to verify the examiners' reliability.

### 2.2. Subgingival Sample Collection

In the DCP group, plaque samples were collected from 3 deepest sites with probing depth of ≥5 mm and in the DH group, plaque samples were obtained from the healthy gingival sulcus around three teeth. All the clinical samples were collected by a Gracey curette under strict asepsis, and their isolation was performed after air-drying and supragingival debridement. The samples were then transferred into a microcentrifuge tube containing transport medium Tris-EDTA buffer (TE buffer) and forwarded to the laboratory for further analysis.

### 2.3. DNA Isolation

The plaque samples were subjected to DNA isolation by the modified proteinase-K method as described previously [16]. In short, the plaque sample was thoroughly vortexed and then washed in TE buffer. The sample was treated with lysis buffer, lysozyme (25 mg/mL) and proteinase K (10 mg/mL) with incubation at 60 °C for 2 h. Proteinase K enzyme was inactivated after 10 min of boiling. The vial was centrifuged, and the supernatant, which contained genomic DNA, was collected in a fresh tube. The DNA was purified by using 3M sodium acetate and absolute ethanol. The purified DNA was aliquoted and stored at −20 °C for further analysis.

### 2.4. Identification of F. alocis and Red Complex Bacteria—PCR Analysis

Polymerase chain reaction was performed by using specific primers for DNA amplification of the 16SrDNA gene as described before by Sequeira et al. (Table 1) [17]. Conventional PCR was used for *F. alocis*, and multiplex PCR for red complex bacterial species. PCR was carried out in a Veriti thermal cycler (Applied Biosystems, Foster city, CA, USA) in 25 μL of reaction mixture (Chromous Biotech, Bengaluru, Karnataka, India) which contained a dNTP mixture (10 mM each), $10 \times$ PCR buffer containing 15 mM $MgCl_2$, Taq DNA polymerase (2.5 units/reaction, and the DNA templates (approximately 100 ng). A primer concentration of 25 pm/μL was used for *F. alocis*, and a concentration of 15 pm/μL was used for each red complex bacterium. The thermal cycler conditions for *F. alocis* were as follows: initial denaturation at 95 °C for 5 min followed by 36 cycles of denaturation at 94 °C for 30 s, annealing at 55 °C for 1 min, extension at 72 °C for 2 min and a final extension at 72 °C for 5 min; for the red complex bacterial species, the conditions were as follows: initial denaturation at 95 °C for 5 min followed by 4 cycles of denaturation at 95 °C for 1 min, annealing at 60 °C for 1 min, extension at 72 °C for 1 min and a final extension at 72 °C for 5 min. The PCR products were further loaded on a 2% agarose gel for electrophoresis and subsequently stained with ethidium bromide (0.5 μg/mL). The gel was photographed under UV light by a gel documentation system (Major Science, Saratoga, CA, USA) [18]. The bands with molecular sizes of 594 bp for *F. alocis*, 404 bp for *P. gingivalis*, 316 bp for *T. denticola* and 641 bp for *T. forsythia* were identified by comparing their position with that of a 100 bp DNA ladder (Figure 1).

**Table 1.** Primer sequences used for the DNA amplification of *F. alocis* and red complex bacteria.

| Organisms | Primer Sequence 5′-3′ | Product Size (Base Pairs) |
|---|---|---|
| *F. alocis* | | |
| Forward | CAGGTGGTTTAACAAGTTAGTGG | 594 |
| Reverse | CTAAGTTGTCCTTAGCTGTCTCG | |
| *P. gingivalis* | | |
| Forward | AGGCAGCTTGCCATACTGCG | 404 |
| Reverse | ACTGTTAGCAACTACCGATGT | |
| *T. denticola* | | |
| Forward | TAATACCGAATGTGCTCATTTACAT | 316 |
| Reverse | TCAAAGAAGCATTCCCTCTTCTTCTTA | |
| *T. forsythia* | | |
| Forward | GCGTATGTAACCTGCCCGCA | 641 |
| Reverse | TGCTTCAGTGTCAGTTATACCT | |

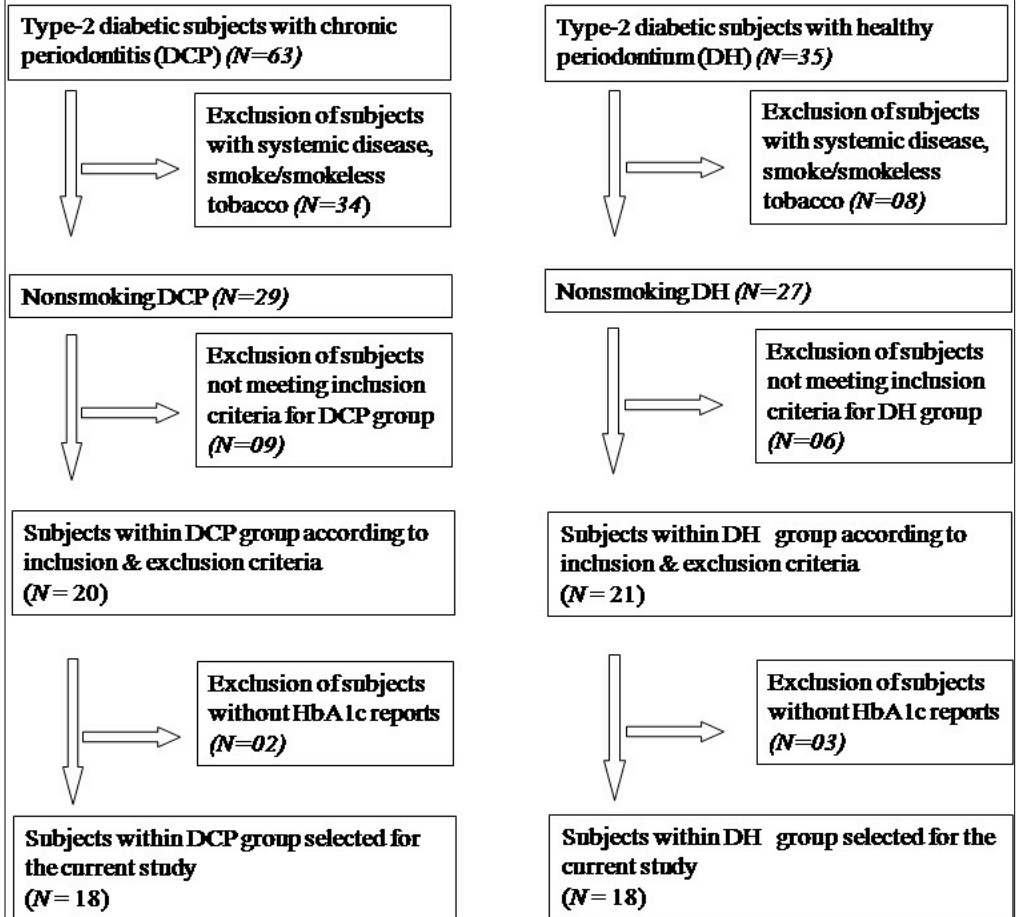

**Figure 1.** Flow diagram for the selection of type 2 DM patients with chronic periodontitis (DCP) and healthy periodontium (DH) according to the inclusion and exclusion criteria.

*2.5. Statistical Analysis*

The mean of the clinical parameters GI, PI, BI, PD, CAL and of the HbA1c values were calculated for each subject, and the average scores were recorded. The detection of *F. alocis* and red complex bacteria in each group and related comparisons were performed using Fisher's Exact Test. The presence of the red complex bacteria with *F. alocis* was analyzed using Fisher's Exact Test. In addition, Mann–Whitney U test and Pearson chi-square test were employed to compare the correlation of the clinical parameters and the HbA1c values

with the presence of *F. alocis*. Any difference with $p < 0.05$ was considered statistically significant. All calculations were performed using SPSS software, version 11.0.

## 3. Results

Of the 98 subjects screened, 63 were diagnosed with generalized chronic periodontitis and were grouped in the chronic periodontitis (DCP group), and 35 control subjects, considered free from periodontitis, were grouped in the healthy periodontium (DH group) [12]. Of the 98 subjects, 34 were excluded because of the use of tobacco in smoking/smokeless form, 9 were excluded based on other exclusion criteria, and 2 participants did not have HbA1c reports. As for the DH group, eight participants were excluded for their history of tobacco use, six subjects were excluded based on other exclusion criteria, and three did not have HbA1c reports. Finally, 18 subjects each in the DCP and DH groups were included in the study. Figure 1 shows a flow diagram for selecting cases and controls based on the inclusion and exclusion criteria specified.

The mean age of the subjects in both the Healthy Diabetic (DH) and the Chronic Periodontitis Diabetic groups (DCP) were $51.2 \pm 8.3$ and $54.02 \pm 9.6$; they included 13 females and 23 males. The detection of *F. alocis* was 11.1% in the DH group and 50% in the DCP group. The comparison between the two groups provided a statistically significant result ($p$ value = 0.02). The frequency of red complex organisms (*T. denticola*, *P. gingivalis* and *T. forsythia*) was analyzed simultaneously, and we found that *T. forsythia* was not detected in the DH group, whereas it was found in 33.33% of the cases in the DCP group. The difference between the groups was statistically significant ($p$ value = 0.01). The odd ratio for *F. alocis*, *P. gingivalis* and *T. forsythia* was found to be less than 1 (0.12, 0.25, 0.051, respectively) (Table 2). The occurrence of red complex bacteria was also compared with the presence or absence of *F. alocis*. This analysis showed a significant copresence of *F. alocis* and *T. forsythia* ($p$ value = 0.006). That is, out of the 11 *F. alocis*-positive subjects, 5 also showed the presence of *T. forsythia* (Table 3, Figure 2). The comparison of the clinical parameters (GI, PI, BI, PD and CAL) in relation to the presence of *F. alocis* in all subjects showed significantly higher values for PD and CAL ($p$ value = 0.01 and $p$ value = 0.03 respectively) (Table 4, Figure 3). It was also noted that good-glycemic-control subjects showed a considerably lower detection of *F. alocis* as compared to fair- and poor-glycemic-control subjects, but the difference was not significant (Table 5, Figure 4).

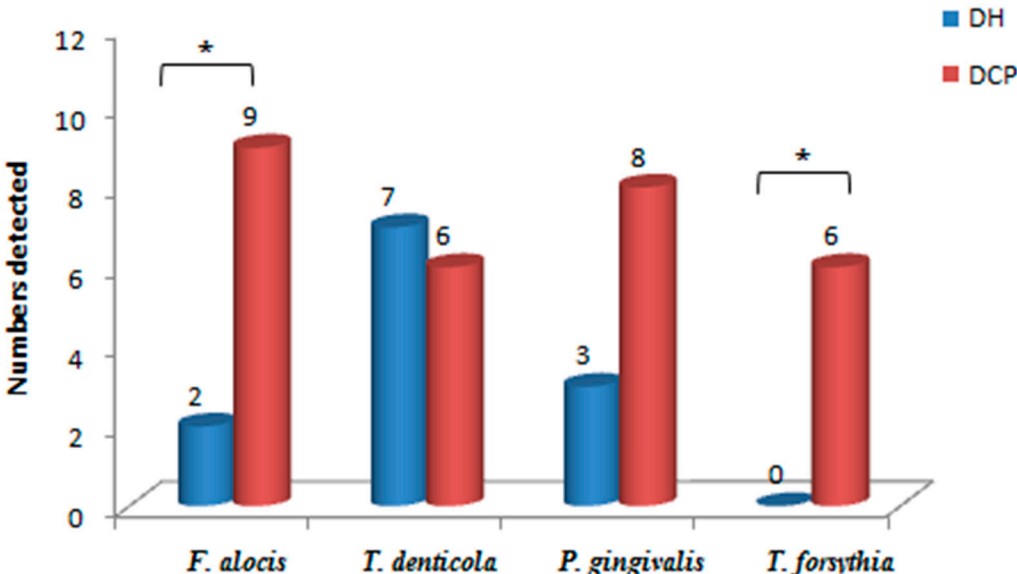

**Figure 2.** Occurrence of *F. alocis* and red complex organisms in the DH and DCP groups. * $p < 0.05$ = significant.

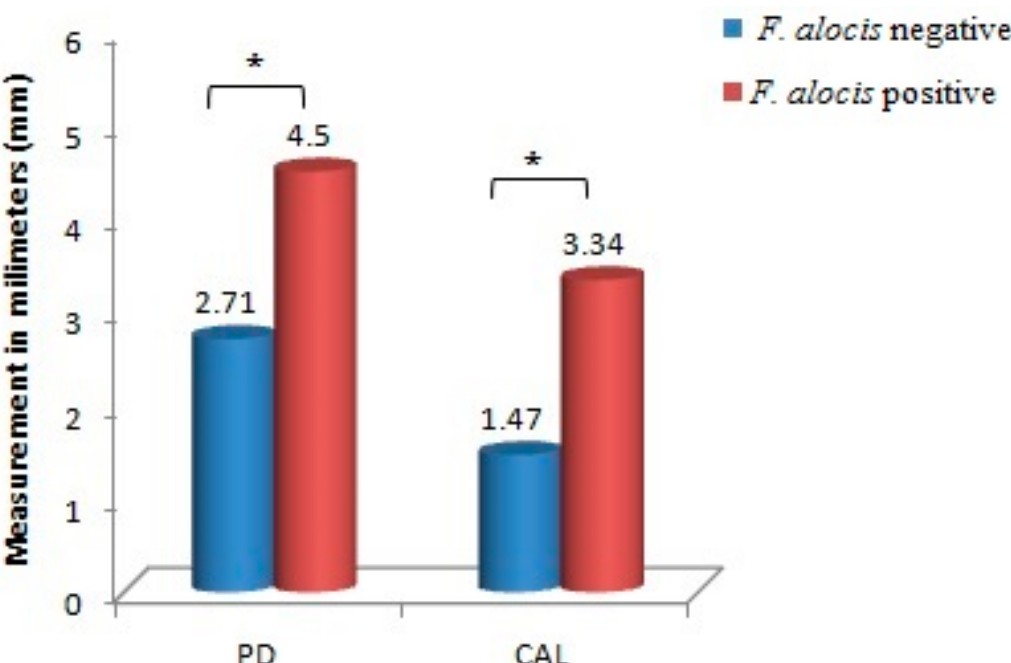

**Figure 3.** Correlation of pocket depth (PD) and clinical attachment loss (CAL) with the presence or absence of *F. alocis*. * *p* < 0.05 = significant.

**Table 2.** Comparison of the two study groups with respect to the occurrence of different bacterial species.

| Organism | DH (%) | DCP (%) | Odds Ratio | 95% Confidence Interval | *p* Value |
|---|---|---|---|---|---|
| *F. alocis* | 2 (11.11) | 9 (50%) | 0.12 | 0.02 to 0.70 | 0.02 * |
| *T. denticola* | 7 (38.88%) | 6 (33.33%) | 1.27 | 0.32 to 4.97 | 1.0 |
| *P. gingivalis* | 3 (16.66%) | 8 (44.44%) | 0.25 | 0.05 to 1.17 | 0.1 |
| *T. forsythia* | 0 (0%) | 6 (33.33%) | 0.051 | 0.002 to 1.00 | 0.01 * |

Fisher exact test, * *p* < 0.05 = Significant; DH, type 2 diabetes mellitus subjects having a healthy periodontium; DCP, type 2 diabetes mellitus subjects having chronic periodontitis.

**Table 3.** Occurrence of red complex bacteria in the presence or absence of *F. alocis*.

| Red Complex Bacteria | | *F. alocis* | | *p*-Value |
|---|---|---|---|---|
| | | Negative | Positive | |
| *T. denticola* | Positive | 9 (36%) | 4 (36.36%) | |
| | Negative | 16 (64%) | 7 (63.64) | 1.0 |
| | Total | 25 (100%) | 11 (100%) | |
| *P. gingivalis* | Positive | 7 (28%) | 4 (36.36%) | |
| | Negative | 18 (72%) | 7 (63.64%) | 0.70 |
| | Total | 25 (100%) | 11 (100%) | |
| *T. forsythia* | Positive | 1 (4%) | 5 (45.45%) | |
| | Negative | 24 (96%) | 6 (54.55%) | **0.006 *** |
| | Total | 25 (100%) | 11 (100%) | |

Fisher's exact Test, * *p* < 0.05 = significant.

**Table 4.** Correlation of clinical parameters with the presence or absence of *F. alocis*.

| Clinical Parameters | *F. alocis* | N | Mean | Standard Deviation | *p*-Value |
|---|---|---|---|---|---|
| GI | Negative | 25 | 0.76 | 0.90 | 0.15 |
| | Positive | 11 | 1.57 | 0.75 | |
| PI | Negative | 25 | 0.78 | 0.92 | 0.08 |
| | Positive | 11 | 1.61 | 0.77 | |
| BI (%) | Negative | 25 | 35.9 | 41.56 | 0.08 |
| | Positive | 11 | 72.55 | 34.69 | |
| PD | Negative | 25 | 2.71 | 1.90 | 0.01 * |
| | Positive | 11 | 4.50 | 1.61 | |
| CAL | Negative | 25 | 1.45 | 2.01 | 0.03 * |
| | Positive | 11 | 3.35 | 1.75 | |

Mann–Whitney U test, * $p < 0.05$ = Significant; N, number; GI, gingival index; PI, plaque index; BI, bleeding index; PD, plaque index; PD, probing depth; CAL, clinical attachment loss.

**Table 5.** Correlation of HbA1c values with the presence or absence of *F. alocis*.

| HbA1$_C$ | *F. alocis* | | *p* Value |
|---|---|---|---|
| | **Negative** | **Positive** | |
| Good diabetic control (6–7%) | 0 / 0% | 1 / 100% | |
| Fair diabetic control (7.1–8.2%) | 11 / 64.7% | 6 / 35.3% | 0.21 |
| Poor diabetic control (>8.2%) | 14 / 77.8% | 4 / 22.2% | |
| Total | 25 / 69.4% | 11 / 30.6% | |

Pearson's chi-square, HbA1c, hemoglobin A1c.

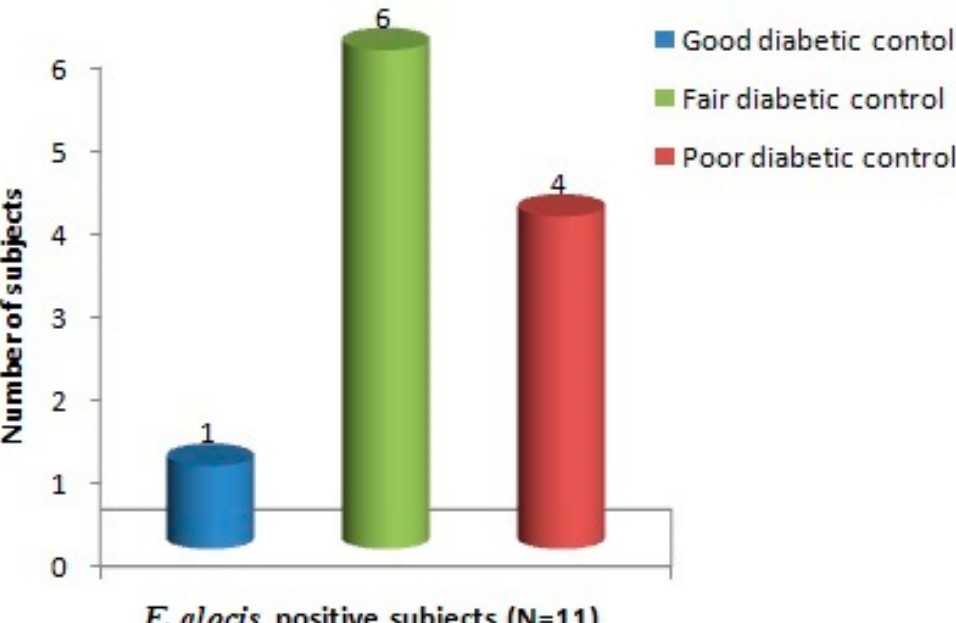

**Figure 4.** HbA1c values and presence of *F. alocis*.

## 4. Discussion

Periodontitis is an infectious inflammatory disease that appears to have an intricate polymicrobial etiology. In particular, the progression of the disease has been linked with the growth of Gram-negative anaerobic species such as *P. gingivalis*, *T. denticola* and *T. forsythia*; however, open-ended molecular techniques such as ribosomal 16S cloning and sequencing have enabled researchers to investigate the microbial profile of any community, including all bacteria in a sample that were previously not suspected and remained uncultivated, including *F. alocis* [19–23]. The PCR assay in this study showed high specificity, and the amplification of all subgingival samples produced all possible results, i.e., positive or negative for all the four target bacteria. PCR detected a significantly greater number of subjects with *F. alocis* in the DCP group as compared to the DH group. This finding is in accordance with studies by various authors that demonstrated *F. alocis* at an increased frequency in periodontal disease sites compared to healthy sites [24–27]. However, this is the first time that the presence of *F. alocis* in CP was evaluated in diabetic subjects. As evidence suggests, hyperglycemia is known to compromise the host response by inhibiting polymorphonuclear neutrophil activity and causing an exaggerated inflammatory immune response to periodontal pathogens. This response induces the release of TNF- α, IL-6 and IL-1β. *F. alocis*, by forming a polymicrobial synergistic relationship, has the ability to cause chronic inflammation via the release of proinflammatory cytokines, which also enhances its invasive capacity [28]. Giugliano et al., in 1995, reported that diabetes patients with high glucose and insulin levels, as well as dyslipidemia, develop microangiopathies, which produce oxidative stress and lead to atherosclerosis [29]. Studies have also shown the resistance of *F. alocis* to oxidative stress and, actually, its stimulated growth under such conditions [2,10]. These conditions could increase the occurrence of *F. alocis* in diabetics and thereby assist in developing periodontal disease. The odd ratio of 0.12 for the negative correlation of *F. alocis* presence with disease in the DCP group suggests that this bacterium was less likely to be detected in the DH group. A similar correlation was found for *P. gingivalis* and *T. forsythia*. As for the red complex bacteria, our study showed that *T. forsythia* was significantly present in a higher number in diabetic patients having chronic periodontitis. This result is in accordance with other previous studies that found *T. forsythia* was associated with poor glycemic control [30,31]. One of the reasons for this association could be the arsenal of virulence factors possessed by *T. forsythia* stimulating the host immune and inflammatory response that leave host cells vulnerable to periodontal infection. According to a theory, the amplitude of the advanced glycation end products (AGE)-mediated cytokine response may be amplified by periodontal infection-mediated cytokine production and secretion. *T. forsythia* was also linked to the body mass index (BMI) due to a significant increase in the depth of the periodontal pocket in obese individuals. Other species of red complex bacteria, on the other hand, did not show this relationship [30].

In 2015, Chen et al. reported a positive correlation of *F. alocis* with *P. gingivalis* and *T. forsythia* isolated from saliva and subgingival samples in periodontitis subjects [32]. Though our study could not find a positive correlation of *F. alocis* with *P. gingivalis* and *T. denticola*, it did so with *T. forsythia*. Both *F. alocis* and *T. forsythia* possess trypsin-like proteases. A study by Haffajee and Socransky in 2009 reported an increased prevalence of *T. forsythia* in overweight or obese individuals compared to normal-weight individuals [33]. Similarly, another study by Brennan et al. in 2007 suggested that, as a result of *T. forsythia* infection, overweight women are considerably more prone to develop periodontitis than normal-weight women [34]. As obesity is a known risk factor for type 2 DM [35], it can be assumed that *F. alocis* and *T. forsythia* could form a synergistic relationship and thus accelerate the disease progression in diabetic subjects. Ardila et al., in 2014, reported a significant association of *P. gingivalis*, *T. denticola* and *T. forsythia* in DCP subjects [36]. Our study could not report similar results, which may be due to our smaller sample size. The results of this study differ from those of previous studies for a variety of reasons, including the methodology utilized, the kind of periodontal disease, and ethnic or geographic characteristics.

In the present study, the results also showed significantly higher PD and CAL in subjects with *F. alocis*. This is in concordance with other previous studies that found this organism in higher frequencies in deeper pockets [10,37,38]. Clinical parameters such as BOP, PD, CAL were also elevated significantly in the presence of other periopathogenic bacteria together with *F. alocis*. This could be attributed to their virulent factors, for instance, resistance, oxidative stress, genes coding for a well-developed amino acid metabolic pathway. Another factor could be the chronic inflammation induced by the release of proinflammatory cytokines that can allow *F. alocis* to colonize and persist in the periodontal pocket's stressful environment with other typical periodontal bacteria [28].

Culture is the gold standard for cultivating bacteria, as it can identify various organisms and their species and can also determine their antibiotic sensitivity. However, culture-based techniques are unable to identify organisms present in low numbers in subgingival samples [39]. PCR, on the other hand, is a very sensitive and specific tool to identify bacteria. It can overcome the difficulties of anaerobic culturing as well as detect the organisms present at low levels and provide rapid results. PCR has the drawback of detecting and amplifying non-viable bacteria and of being invariably expensive [39]. Quite a few studies around the world have evaluated the occurrence of *F. alocis* in CP subjects, but this is the first study that we are aware of that was conducted on Indian subjects, evaluating the occurrence of *F. alocis* in individuals having Type 2 DM. As a result, there is a scarcity of comparable data. Our research was conducted on a tiny fraction of the Indian population; thus, the results may not be representative of the entire subcontinent. The results could not be obtained for the DCP and DH groups for all analyses and are presented here as exploratory data. The present study demonstrated that subjects with good glycemic control hardly showed the presence of *F. alocis*. As already established, the microflora at periodontal disease sites in diabetics is different from that in diseased sites in non-diabetics, which may be because of environmental factors affecting the subgingival ecosystem [9]. Non-diabetics and diabetics show robust and distinctive microbial centers, and *F. alocis* was identified in the diabetic microbial hub [9]. The susceptibility and severity of periodontal disease in diabetics may also be due to a problematic host response which includes an exaggerated inflammatory response and oxidative stress [40]. A good glycemic control might ameliorate these issues and hence may inhibit the occurrence of putative pathogenic species such as *F. alocis*.

## 5. Conclusions

In conclusion, a subset of the Indian population having type 2 DM was evaluated in the present study for the presence of *F. alocis*, for the first time. The findings suggest an increased prevalence of this bacterium in type 2 diabetic subjects having chronic periodontitis. In addition, they showed a positive correlation of *F. alocis* presence with increased PD and CAL. Finally, the synergistic combination of *F. alocis* with *T. forsythia* may lead to increased severity of the periodontal disease in diabetic subjects. Further studies with a larger sample size, salivary glucose levels and normoglycemic groups with a healthy periodontium and chronic periodontitis would help provide a complete picture of the occurrence of this organism and its correlation with the red complex.

**Author Contributions:** Conceptualization, K.G.B. and V.M.J.; methodology, H.F.M.S., P.U.O. and M.S.K.; writing—original draft preparation, S.S.K. and V.M.J.; writing—review and editing, M.S.K. and V.M.J. All authors have read and agreed to the published version of the manuscript.

**Funding:** This research received no external funding.

**Institutional Review Board Statement:** The study was approved by the Institutional Review Board of MARATHA MANDAL'S NGH INSTITUTE OF DENTAL SCIENCES AND RESEARCH CENTRE, BELGAVI (Certificate no. 2015-16/1118).

**Informed Consent Statement:** Informed consent was obtained from all subjects involved in the study.

**Data Availability Statement:** The data presented in this study are available on request from the corresponding author.

**Acknowledgments:** The authors would like to thank Ravi Shirhatti, Department of Community Dentistry and biostatistician, SDM College of Dental Sciences and Hospital, Karnataka, India, for his help in the statistical analysis.

**Conflicts of Interest:** The authors declare no conflict of interest.

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
