# Peer review of "Co-Occurrence of Filifactor alocis with Red Complex Bacteria in Type 2 Diabetes Mellitus Subjects with and without Chronic Periodontitis: A Pilot Study"

_2673-8937, doi:10.3390/ijtm3010009_

Round 1
Reviewer 1 Report
The aim of this study was to investigate the occurrence of Filifactor alocis in Type 2 diabetes patients with chronic periodontitis compared with Type 2 diabetics without periodontitis. The study used PCR to determine presence of F. alocis as well as the red complex bacteria and found a correlation between the presence of F. alocis and chronic periodontitis including increased PD and CAL. The study has some nice data but the presentation is poor and needs work.
There are a large number of minor editing/grammatical errors – too many to list them all. Eg line 52 has the title of reference 9 included. Gram (line 54) should be capitalized as it is someone’s name. All of the references to a temperature are missing the superscript 0 eg. line 119 has temperature of 600C. Line 123 refers to -200C. Species names should always be in italics. Should co-relation be correlation or do you mean co-occurrence?
The introduction should include more information about HbAc1 so the importance of it as an indication of how well diabetes is being controlled is evident. Otherwise it is fine.
Materials and methods
- This section needs work. Lines 67-77 contain information that should be in the results. The methods should state that patients with Type 2 diabetes reporting to the outpatient department were recruited and a systematic methodology was used to interview them. They were divided into those with chronic periodontitis or healthy periodontium. Be more precise in the description of the inclusion and exclusion criteria. Inclusion – Type 2 diabetes, more than 20 teeth, HbA1c of more than 6.0. Exclusion – use of tobacco, less than 20 teeth, no HbAc1 reading, criteria outlined in lines 90-94. Also keep the description of how patients were separated into chronic periodontitis or healthy periodontium.
- The rest of the information belongs in the results section, including Figure 1.
- Figure 2 also belongs in the results and the figure legend needs to be expanded to explain what is being shown in the picture. What samples are in lanes 1-4? Are they representative of one particular group?
- Need more information about the clinical parameters. They were used to place patients in groups so how were they assigned? eg. What plaque index number, what probing depth, etc was needed for patients to be in the DCP group?
-
Results
Should include the information that has been removed from the methods section.
Consider improving the organisation of the Tables and Figures as they are hard to follow. Place the Figures near the Tables if you wish to keep them all but it is not necessary to have them both as there seems considerable overlap. The Tables are more informative but it would be helpful if the formatting were changed to make it easier to read. ie more lines. The Figure legends are too brief so would need expanding if the Figures are to be kept. It is very difficult to work out what the data is that has been included. Is it for the whole group or just the DCP or just the DH group?
Also pay attention to the axes. Figure 4 has subjects with red complex bacteria on the y axis and the increments are 0.5. Presumably you can only have whole numbers if it refers to patients.
Is it necessary to have mean and median for the measures in Table 4? It complicates the analysis of the Table. I would also consider leaving out total columns in the relevant Tables as they don’t add anything but make the Table harder to understand.
Did any of the patients have more than one of the red complex bacteria as well as F. alocis? It would be interesting to see if patients with multiple species had deeper pockets and higher CAL measurements.
Discussion
The discussion is okay but needs rearranging. You should be highlighting the connection between the presence of the bacteria and the likelihood of having chronic periodontitis at the start of the discussion. The limitations of using a small number of patients in a small area of India should be moved towards the end of the discussion.
Line 257-260 refers to a significantly greater levels of F. alocis in DCP group compared with DH group. Do you mean a significantly greater number of patients have F. alocis? The PCR you have done doesn’t permit quantification.
Check the references as there seems to be double numbering occurring.
Author Response
Thank you for reviewing our paper and for all the suggestions provided. We have addressed the issues that was pointed out.

Reviewer 2 Report
1. This paper is a highly original topic and I read it with great interest. However, while this study appears to be a case-control study that examined patients with and without periodontitis, many of the results (Tables 3-5, Figures 3-6) of the analysis are not related to the presence or absence of periodontitis. If the study design is case-control, then it is uncomfortable to analyze data from patients with periodontal disease and gingivally healthy subjects together.
2. Did you calculate the sample size of study population before conducting the study?
3. Please clarify the recruitment period of study population.
4. In this study, patients with HbA1c ≥ 6.0 were selected for the study (lines 84-85). There appear to have a mix of those with good and poor glycemic control, what is the rationale behind your decision for a value of 6.0?
5. In the DCP group subjects that demonstrated presence of gingival inflammation with bleeding on probing, PD ≥ 5mm and CAL ≥ 3mm were included (lines 88-89). Please cite the reference.
6. The clinical periodontal examination comprised the assessment of the plaque index …two examiners performed all measurements with a millimeter-graded, probe (UNC 15) (lines 102-106). Am I correct in understanding that the researcher who conducted the dental check-up also took blood samples and measured HbA1c?
7. In the DH group, plaque samples were obtained from healthy gingival sulcus (lines 110-111). From which tooth gingival sulcus was the sample taken?
8. Table 2, How was the odds ratio obtained? Not stated in the method section.
Author Response
Thank you reviewing our paper and for the suggestions made. We have addressed all the concerns that has been pointed out.

Reviewer 3 Report
Dear Editor,
thank you for your request to review the manuscript “Co-occurrence of Filifactor alocis with red complex bacteria in type 2 diabetes mellitus subjects with and without chronic periodontitis”.
The article is interesting, and the paper should be accepted after making a revision.
I have some comments:
in the
- In the introduction section, in line 57 the Authors mentioned “previous studies” but the cited only one; they should add all the studies considered
- In the methods the Authors wrote that the patients were quizzed about their medical history and medication but they do not specify in the text or in tables this parameter. Do they consider these variables?
- In the exclusion criteria the Authors wrote they have excluded patients with “antimicrobial therapy in previous 3months before sampling” ; they do not specify if they refer to the topical or systemic antimicrobial therapy.
- The Authors do not discuss the impact of the salivary glucose levels on the periodontitis and bacteria proliferation.
- The Authors should add a limitation section including that they did not evaluate the salivary glucose; they did not evaluate a normoglycemic groups (moving this from the conclusion);
Author Response

(The authors gave the same response as above.)

Round 2
Reviewer 1 Report
The aim of the study was to evaluate prevalence of F. alocis in T2DM patients with and without chronic periodontitis. Also wanted to see how many of each group had red complex bacteria as well as F. alocis.
This is a revised version of a previous submission and it has improved but I still have some concerns about the presentation of the manuscript. The results start off by separating the patients into the two groups as described and evaluating presence or absence of the bacteria in the two groups. This is well presented (Table 2 and Figure 3). The next few figures and tables are confusing as the groups are combined and the data is much harder to interpret. It is not clear in Table 3 which patients belong to DCP and which to DH although this information is in the associated figure. My suggestion would be to remove the Table and keep the figure as it is easier to interpret.
The text (lines 195-198) talks about differences between the groups (DCP and DH) but there doesn’t appear to be that separation in Table 4 or in figure 5. This would be interesting information. There is a clear difference in PD and CAL between F. alocis positive and negative patients and it would be good to know which group they belonged to.
Does Fig. 6 only include patients who were F. alocis positive and DCP? There is not enough information in the figure legend to be able to work this out.
Figure 1 – It is still not clear what template DNA was used for the PCR reactions shown. Did you have DNA isolated from known bacterial samples or did you use samples from patients included in the study? If not using DNA from known bacteria were any of the bands sequenced to confirm they were correctly amplifying the specified bacteria? If you have used samples from patients it belongs in the results section. It is not necessary to include all samples from all patients. You can say they are representative samples. I would suggest removing the figure altogether if you do not wish to place it in the results.
The discussion is still in need of rearrangement. The focus of the start should be about their findings and placing them in the context of the broader literature not be talking about drawbacks of PCR. That can go further down. Some of that info should be in methods eg. Lines 274-276 or intro. Eg lines 263-268
Minor points: Lines 36-37 – it is not necessary to shorten the species name in brackets. It is enough to use the full name the first time then shorten on subsequent uses.
Some weird phrasing in some sentences.
Line 83 “Type 2 DM subjects were selected …” is a redundant sentence.
Line 188, 301, 303 T. forsythia not italicized
Table 3 (line 211) still has totals in it.
Table 4 heading should be correlation not co-relation
Table 2 and Figure 3 are placed far apart but should be closer together.
Figure legends need more details.
Line 254 Gram needs to be capitalized.
Line 288 don’t use PMN abbreviation
Line 290 – misleading as it suggests that F. alocis is releasing IL-1β, IL-6 and TNF.
Author Response
Thank for your reviewing our paper and suggesting the changes. The suggested changes are made and answers to comments attached.

Reviewer 2 Report
The authors fully responded to my concerns.
Author Response
Thank you for reviewing our paper.
Reviewer 3 Report
The author addressed all the concerns
Author Response
Thank you for your review.
Regards
Round 3
Reviewer 1 Report
The stated aim of this study was to look at the prevalence of F. alocis in T2DM patients with chronic periodontitis compared with the prevalence in T2DM with healthy periodontium. A substantial part of the paper is devoted to describing how patients were sorted into those with CP and those without. Figure 2 shows that more patients with CP have F. alocis than patients without CP. The remaining figures then combine the groups to look at some other parameters. Why bother to separate them out into DCP and DH if you are not going to keep them separate for all of the parameters looked at in the paper? I know the numbers are small and statistical analysis is difficult but the better story is to be found in keeping the groups separate for all of the analyses.
Author Response
Response to Reviewer 1 Comments
The stated aim of this study was to look at the prevalence of F. alocis in T2DM patients with chronic periodontitis compared with the prevalence in T2DM with healthy periodontium. A substantial part of the paper is devoted to describing how patients were sorted into those with CP and those without. Figure 2 shows that more patients with CP have F. alocis than patients without CP. The remaining figures then combine the groups to look at some other parameters. Why bother to separate them out into DCP and DH if you are not going to keep them separate for all of the parameters looked at in the paper? I know the numbers are small and statistical analysis is difficult but the better story is to be found in keeping the groups separate for all of the analyses.
Response: For Figure 2; the analysis between the DCP and DH group was possible as presence of single organism (F. alocis or either of the red complex bacteria) was looked into and the numbers available was suitable for statistical analysis using the specified test. When the occurrence of either of the red complex bacteria with F. alocis was looked into the numbers occurring in either group was very limited, so the groups were combined to check if any combination of red complex bacteria with F. alocis stood out, which has been shown in Table 3.
For the clinical parameters Table 4, Intra group comparison for F. alocis presence/ absence for PD and CAL and other parameters did not show any statistical difference, but for inter group comparison there will be obvious difference in all the parameters between the groups, merely based on the selection criteria for each group. and for this very reason we combined the groups to analyze (Table 4 & Figure 3). The reason for combining the groups was to eliminate that obvious difference between the groups and demonstrate the fact that F. alocis presence was related to deeper PD or higher CAL.
Table 5 and Figure 4: The groups were combined for this analysis was to check the difference in the occurrence of F. alocis based on the diabetic control. The statistical analysis was not possible even with the two groups combined.
The results helped us check our hypothesis and have been presented as an exploratory data for further long term and large sample-based studies. This point has been now mentioned in the discussion as well.